# The ‘Shape-Shifter’ Peptide from the Disulphide Isomerase PmScsC Shows Context-Dependent Conformational Preferences

**DOI:** 10.3390/biom11050642

**Published:** 2021-04-26

**Authors:** Lorna J. Smith, Chloe W. Green, Christina Redfield

**Affiliations:** 1Department of Chemistry, University of Oxford, Oxford OX1 3QR, UK; greenchloe831@gmail.com; 2Department of Biochemistry, University of Oxford, Oxford OX1 3QU, UK

**Keywords:** NMR spectroscopy, MD simulation, X-ray crystallography, protein dynamics, chameleon sequence, protein folding, protein misfolding, intrinsically disordered protein, molecular recognition features

## Abstract

Multiple crystal structures of the homo-trimeric protein disulphide isomerase PmScsC reveal that the peptide which links the trimerization stalk and catalytic domain can adopt helical, β-strand and loop conformations. This region has been called a ‘shape-shifter’ peptide. Characterisation of this peptide using NMR experiments and MD simulations has shown that it is essentially disordered in solution. Analysis of the PmScsC crystal structures identifies the role of intermolecular contacts, within an assembly of protein molecules, in stabilising the different linker peptide conformations. These context-dependent conformational properties may be important functionally, allowing for the binding and disulphide shuffling of a variety of protein substrates to PmScsC. They also have a relevance for our understanding of protein aggregation and misfolding showing how intermolecular quaternary interactions can lead to β-sheet formation by a sequence that in other contexts adopts a helical structure. This ‘shape-shifting’ peptide region within PmScsC is reminiscent of one-to-many molecular recognition features (MoRFs) found in intrinsically disordered proteins which are able to adopt different conformations when they fold upon binding to their protein partners.

## 1. Introduction

Chameleon sequences, amino acid sequences that adopt different secondary structure conformations in different protein structures, are currently attracting considerable interest [1]. Studies of these sequences in different contexts firstly provides an understanding of the fundamental determinants of protein structure and dynamics [2,3]. Secondly, the properties of these sequences can give insights into protein folding diseases, such as Alzheimer’s and transmissible spongiform encephalopathies, where a native protein structure undergoes a major conformational change to form amyloid fibrils [4,5]. Finally, these sequences have the potential to be utilized as tools in protein engineering and design, for example in stimulus-responsive peptide systems which undergo a conformational transition prompted by changes in the external environment [6]. Chameleon sequences are also relevant in intrinsically disordered proteins (IDPs) in the context of short molecular recognition features (MoRFs) which can adopt different conformations upon binding to different partner proteins [7,8,9,10].

Recently, a so called ‘shape-shifter’ peptide, has been identified in *Proteus mirabilis* suppressor of copper sensitivity protein C (PmScsC), a homo-trimeric protein disulphide isomerase [11]. Each monomer unit consists of an N-terminal trimerization stalk and a thioredoxin-fold catalytic domain linked by an eleven-residue peptide motif. Three crystal structures of PmScsC have been determined under different crystallization conditions: compact, transitional and extended [11]. The compact and transitional crystals were obtained using a 2.85 M sodium malonate well solution at pH 5.8 while the extended crystals were obtained using a 0.1 M HEPES well solution at pH 8. The structures of the trimerization stalk and catalytic domains are essentially unchanged between these three forms but the linker peptide adopts a helical, strand or loop conformation resulting in significant changes in the quaternary structure of the trimer. In the extended structure the linker is helical and the catalytic domain is rotated and positioned away from the trimerization domain (Figure 1a) while in the compact form the catalytic domains are positioned close to the trimerization stalk and the linker forms a connecting loop (Figure 1b). The transitional crystal structure contains monomers adopting both the compact and extended conformations along with an intermediate conformation in which the linker forms a short β-strand hydrogen bonding to the catalytic domain positioned above the trimerization stalk (Figure 1c). The eleven-residue linker peptide has been called a ‘shape-shifter’ peptide based on its ability to adopt different secondary structures and to influence the relative orientations of the domains that it links [11].

A variant of PmScsC in which the eleven-residue linker is deleted, PmScsCΔLinker, lacks disulphide isomerase and dithiol oxidase activities [12]. In this variant, a mixture of monomer, dimer and trimer are present in solution, whereas only trimer is observed for the wild-type protein. A crystal structure of the PmScsCΔLinker trimer shows that the three catalytic domains are packed closely together. This suggests that the linker peptide plays an important role as a spacer between the trimerization stalk and catalytic domains which enables a stable and functional trimer to form. Replacement of the linker in PmScsC by a rigid helical peptide also leads to the loss of disulphide isomerase activity showing that dynamical motion is essential for the mechanism of PmScsC [11].

In light of the important role of the shape-shifter linker peptide in the function of PmScsC and its interesting conformational properties, we have characterised this linker peptide in solution. In particular, experimental NMR studies and molecular dynamics (MD) simulations have been used to study a peptide with a sequence corresponding to residues 38–50 (the eleven-residue linker and one additional residue at each end) of PmScsC. The X-ray structures of PmScsC have also been analysed to identify intermolecular interactions involving the linker peptide sequence. The results highlight the important role that quaternary interactions can play in determining conformational properties.

## 2. Materials and Methods

### 2.1. Sample Preparation

The thirteen-residue linker peptide, with sequence KKADEQQAQFRQA, was purchased from Pepceuticals Limited (Leicester, England). The N- and C-termini were modified with acetyl and amide groups, respectively. Peptide samples for NMR were 2.5 mM in either 0.01 M sodium phosphate buffer (95% H_2_O/5% D_2_O) at pH 7 or a 50:50 mixture of deuterated trifluoroethanol (TFE) and 0.01 M sodium phosphate buffer at pH 7. A small amount of 2,2-dimethyl-2-silapentane-5-sulfonic acid (DSS) was added as a chemical shift reference.

### 2.2. NMR Spectroscopy and NMR Data Analysis

NMR experiments were carried out at 288 K using 500, 600 and 950 MHz spectrometers equipped with Bruker Avance consoles. The 500 and 600 MHz spectrometers were equipped with TCI CryoProbes and the 950 MHz spectrometer with a room temperature probe.

Resonance assignments for the peptide in H_2_O and in 50% TFE were obtained using 2D ^1^H-^1^H COSY, TOCSY and NOESY experiments and natural abundance ^1^H-^13^C HSQC and ^1^H-^15^N HSQC experiments. Data for the peptide in 95% H_2_O/5% D_2_O were collected at 600 MHz with sweep widths of 6024.096 Hz, 9900.99 Hz and 1524.390 Hz in the ^1^H, ^13^C and ^15^N dimensions, respectively. Additional TOCSY and NOESY experiments collected at 950 MHz used a ^1^H sweep width of 9615.385 Hz. Data for the peptide in 50% TFE were collected at 500 MHz with sweep widths of 5376.344 Hz, 8264.463 Hz and 1267.427 Hz in the ^1^H, ^13^C and ^15^N dimensions, respectively. 1D spectra used to measure ^3^J_HNHα_ coupling constants were collected at 950 MHz.

NMR data were processed using NMRPipe [13] and analysed using CcpNmr Analysis [14]. Spin system information was obtained from analysis of the COSY, TOCSY and ^1^H-^13^C HSQC spectra; the latter contained both one-bond and two-bond ^1^H-^13^C correlations. Sequence specific assignments were obtained from analysis of HN(i)-HN(i+1), Hα(i)-HN(i+1) and Hβ(i)-HN(i+1) NOEs. ^1^H and ^13^C chemical shifts were referenced using DSS, and ^15^N chemical shifts were referenced indirectly. The resonance assignments in water and in 50% TFE have been deposited in the BMRB with deposition codes 50399 and 50400, respectively.

Analysis of NMR chemical shifts to obtain secondary structure predictions was carried out using the Secondary Structure Propensities (SSP) software [15]. The SSP scores were calculated with the default averaging window of 5 residues but also without averaging (m = 1). The ΔC^α^-ΔC^β^ and ΔH^α^ secondary shifts were also calculated using SSP. All structural figures were rendered using PyMOL [16].

### 2.3. MD Simulations

Molecular dynamics simulations were performed using the GROMOS bio-molecular simulation software [17,18] with the GROMOS 54A7 force field [19]. Four 100 ns MD simulations of the linker peptide were performed and one 20 ns MD simulation of the protein trimer. The initial structures for the peptide simulations were taken from residues 38–50 in the X-ray structures of PmScsC [11] with protein data bank codes 4xvw (molecule A; loop), 5id4 (helix) and 5idr (molecules A; irregular helix and B; β-strand). Note that there is a regular pattern of NH(i)–CO(i-4) hydrogen bonds along the length of the peptide in the 5id4 structure but in the 5idr molecule A structure the hydrogen bonding pattern seen is not that of a regular α-helix with four missing NH(i)–CO(i-4) hydrogen bonds in the centre of the peptide and one NH(i)–CO(i-3) hydrogen bond characteristic of a 3_10_ helix (45 NH-42 O). The initial structure for the protein simulation was the 5id4 X-ray structure. The peptide or protein molecules were solvated in rectangular boxes and minimum image periodic boundary conditions were applied. The minimum solute-box wall distance was set to values ranging from 0.9 to 1.4 nm depending upon the starting conformation, yielding 3647 (4xvw and 5id4 peptide simulations), 4996 (5idr molecule B peptide simulation), 5256 (5idr molecule A peptide simulation) and 58,785 (5id4 protein simulation) SPC water molecules [20]. To compensate for the overall positive charge of the solute, one and three Cl^-^ ions were included in the peptide and protein simulations, respectively.

For each of the simulations an equilibration scheme comprising five 20 ps simulations of 60 K, 120 K, 180 K, 240 K and 298 K was used at constant volume. During the first 80 ps of this equilibration the solute atoms were harmonically restrained to their positions in the initial structures with force constants of 25,000, 2500, 250 and 25 kJ mol^−1^ nm^−2^ at temperatures of 60 K, 120 K, 180 K and 240 K, respectively. Following this equilibration, all simulations were performed at a temperature of 298 K and a pressure of 1 atm using the weak-coupling algorithm [21], with relaxation times of τ_T_ = 0.1 ps and τ_p_ = 0.5 ps and an isothermal compressibility of 4.575 × 10^−4^ (kJ mol^−1^ nm^−3^)^−1^. Solute and solvent were separately coupled to the heat bath. The SHAKE algorithm [22] was used to constrain bond lengths of the solutes and the rigid geometry of the solvent molecules, with a relative geometric tolerance of 10^−4^ allowing for an integration time step of 2 fs. The centre of mass motion of the system was removed every 1000 time steps. Non-bonded interactions were calculated using a triple-range cut-off scheme with cut-off radii of 0.8 nm and 1.4 nm. Interactions within 0.8 nm were evaluated every time step and intermediate interactions were updated every fifth time step. To account for the influence of the dielectric medium outside the 1.4 nm cut-off sphere, a reaction-field force [23] with a dielectric permittivity ε_RF_ = 61 for water was used. The solute configurations were saved for analysis every 5 ps.

In the analysis of the simulation trajectories hydrogen bonds were identified according to a geometric criterion: a hydrogen bond was assumed to exist if the hydrogen-acceptor distance was smaller than 0.25 nm and the donor-hydrogen-acceptor angle was larger than 135°. Conformational clustering was performed using the algorithm of Daura et al. [24] and the backbone N, CA and C atoms of residues 3–11. The atom positional RMSD cut off used to determine the structures belonging to a single cluster was 0.15 nm. Trajectory structures lying 10 ps apart were used.

## 3. Results

2D ^1^H-^1^H COSY, TOCSY, NOESY and natural abundance ^1^H-^13^C and ^1^H-^15^N HSQC experiments were used to assign the spectrum of the linker peptide in aqueous solution at pH 7 (Appendix A). The secondary chemical shifts for the peptide under these conditions were all small (ΔC^α^-ΔC^β^ in the range −0.76 to +0.55 ppm and ΔH^α^ in the range −0.06 to +0.07 ppm) (Figure 2a,b). The chemical shift data were analysed using the Secondary Structure Propensities (SSP) algorithm [15]. All residues gave low SSP scores in the range of +/− 0.15 when applying the default 5-residue averaging window (Figure 2c). Without this averaging, slightly larger SSP scores are obtained with a small preference for extended structure at the N- and C-termini and a small preference for helix at residues Asp 41, Glu 42, Ala 45 and Gln 46. Overall, the SSP scores of ~0.10 indicate that only ~10% of the conformational ensemble populated would be adopting a regular helical conformation. The unstructured, random coil nature of the linker peptide in aqueous solution was further confirmed by the observation of only intraresidue and sequential NOE’s and by ^3^J_HNα_ coupling constants in the range 6.7–8.1 Hz (Figure 2d). Predictions from Agadir [25] for the sequence were in good agreement with these results, predicting approximately 3% overall helicity for the peptide in aqueous solution.

The peptide was also characterised in trifluoroethanol (TFE) solution. A titration monitored using CD spectroscopy showed no cooperative transition on the addition of TFE to the peptide solution, and 50% TFE (*v*/*v*) was needed to induce a significant helical population (Appendix A). This is in contrast to peptides which have a considerable helical propensity in aqueous solution, where often significant increases in helicity are seen on the addition of low concentrations of TFE [27]. NMR studies were performed in 50% TFE solution (*v*/*v*) and the secondary chemical shifts ΔC^α^-ΔC^β^ were in the range 1.6–2.7 ppm for residues 40–46 (Figure 2a). Analysis of the chemical shift data using the SSP approach [15] gave scores in the range 0.42–0.74 for residues 40–46 meaning that approximately half the ensemble of conformers populated adopt a helical conformation (Figure 2c). The residues towards each terminus have lower SSP scores in the range −0.14 to +0.30. The helical propensity in 50% TFE solution was confirmed by the observation of H^α^(i)-H^β^(i+3) and H^α^(i)-H^N^(i+3) NOE’s along the peptide sequence. The ^3^J_HNα_ coupling constant values measured under these conditions were in the range 4.2–6.8 Hz (Figure 2d). In general, the measured ^3^J_HNα_ was larger than the value of 3.9 Hz typically seen for a regular helix in a protein but for residues 40–46 the J value was below the value predicted for a random coil. The NOE and coupling constant data are consistent with the estimate of ~50% helical conformers from SSP.

The NMR studies in solution have been complemented by molecular dynamics (MD) simulations. Firstly, four 100 ns simulations were run for the linker peptide (residues 38–50) in aqueous solution using starting coordinate sets from the different structures the peptide adopts in the reported X-ray structures (Figure 3, column in the centre): loop from the compact structure, regular helix from the extended structure, and β-strand and irregular helix from the transitional structure. The hydrogen bonds present in the starting coordinate sets are summarised in Table 1 and the Φ, Ψ torsion angles are shown in Appendix A.

In all four of the MD simulations run, the linker peptide showed significant deviations from its starting structure, adopted a wide range of conformations and sampled a number of different hydrogen bonds during the simulations (Table 1). The peptide conformations in the four 100 ns trajectories have been taken together (40,000 structures) and conformational clustering was performed (see Methods Section 2.3). Seven main clusters were identified to which 75% of the peptide structures belong (Figure 3 and Appendix A). The two largest clusters each contained approximately 20% of the peptide structures. In cluster 1 the peptide adopts a distorted hairpin type structure, which enables formation of a 40NH-49O hydrogen bond (92% population) while in cluster 2 the structures show a high population of a helical hydrogen bond in the centre of the peptide (45NH-41O 89%,) The structures in clusters 1 and 2 are derived almost exclusively from the MD trajectories starting from the loop and irregular helix structures, respectively. Clusters 3 and 5 contain structures which mostly have helical hydrogen bonds only in the middle or at the N-terminus of the peptide, respectively. Structures in these clusters result from several different trajectories (Figure 3, Appendix A). In contrast, cluster 4 contains structures which have helical hydrogen bonds along the length of the linker peptide. This cluster contained 11% of the 40,000 structures analysed and hence is similar to the level of helicity suggested by the SSP score for the peptide in aqueous solution. It is interesting to note that ~50% of structures in cluster 4 result from the MD trajectory that starts from the β-strand peptide conformation (Figure 3, Appendix A). Except for the structures in cluster 4, where in general the Φ, Ψ torsion angles are typical of a regular helix, residues with Φ, Ψ torsion angles in both the α and β regions of the Ramachandran plot are seen, although hydrogen bonds characteristic of a β-strand conformation cannot form in such a short peptide fragment alone.

To gain further understanding into the conformations of the peptide seen in the different crystal structures of PmScsC, the intermolecular contacts, hydrogen bonds and salt bridges were analysed, both within the trimer and between other molecules in the asymmetric unit, using the Protein Interfaces, Surfaces and Assemblies service (PISA) [28] and the Evolutionary, Protein–Protein Interface Classifier (EPPIC) webserver [29]. In all the crystal structures, the N-terminal region of the three molecules in the trimer come together to form an irregular three-helix coiled coil with hydrogen bonds and salt bridges between the chains involving the side chains of Glu 12, Arg 14, Arg 18, Asp 19, Glu 28 and Glu 30. The linker peptide sequence follows this trimerization stalk. In the extended crystal structure, the only hydrogen bonds or salt bridges involving the linker peptide region are at the N-terminal end from the side chains of Lys 38 and Lys 39 to the side chains of Glu 26 and Glu 29 in the trimerization stalk of a neighbouring subunit in the trimer (Figure 1a). In the compact crystal structure there are contacts between the linker peptide sequence and residues 116–123 and 151–153 in the catalytic domain of a neighbouring subunit within the trimer with hydrogen bonds between the side chains of Asp 41 and Gln 44 on one subunit and Lys 120 and Asn 152 on the neighbouring subunit (Figure 1b).

The network of contacts formed by the linker sequence in the crystal structure of the transitional form are more complex as they are dependent on the conformation of the linker peptide. For molecule A, where the linker peptide forms an irregular helix, there are contacts to molecule D which is in a neighbouring trimer. These particularly involve hydrogen bonds and salt bridges between Lys 38, Glu 42 and Gln 46 on chain A and Glu 182, Arg 185 and Lys 186 on chain D (Figure 1d). For molecule B, where the linker peptide forms an extended β-strand, there are intermolecular contacts from the linker peptide to residues 25–44 and 200–217 in molecule A within the trimer. Here, hydrogen bonds are formed between Gln 44 and Arg 48 in chain B to Gln 36 and Glu 217 in chain A (Figure 1c). Finally, the linker peptide in molecule C forms a loop and the coordinates of residues 46–47 are absent in the crystal structure. Here, the linker peptide sequence forms contacts with residues 117–120 of molecule D in a neighbouring trimer, with a salt bridge between Asp 41 in chain C and Lys 120 in chain D (Figure 1e).

Under solution conditions, any hydrogen bonds between molecules in different trimers will clearly be missing. In addition, any intramolecular or inter-monomer hydrogen bonds involving side chain groups on the surface of the protein are expected to be fluctuating with short lifetimes [30]. Consequently, the quaternary structure of the trimer is likely to be an ensemble of conformers, rather than any one of the different structures seen in crystals predominating in solution. An indication of this was seen in a short 20 ns simulation of the trimer of the full PmScsC protein run starting from the extended structure seen in the 5id4 crystal structure (Figure 1a). Here, all three linker peptides started in a regular helical conformation. However, even in the short simulation time the N-terminal region of the linker peptide in molecule B started to unfold with residues 38–40 adopting extended β-strand like conformations (Figure 4), and a number of short-lived hydrogen bonds involving side chains of residues in the linker region being observed. In addition, the quaternary structure of the trimer became much more compact with the catalytic domains moving towards each other to fill space that was occupied by symmetry related molecules in the crystal structure.

## 4. Discussion

The linker peptide in PmScsC was initially identified as a shape-shifting motif which had potential for plug-and-play applications in protein engineering [11]. The studies of this peptide sequence alone in solution reported here show that it is essentially disordered and the different conformations it adopts in crystal structures of PmScsC are a consequence of different intermolecular hydrogen bonding patterns and crystal contacts formed under a variety of crystallization conditions which differ in pH and ionic strength [11]. The compact and transitional crystals were obtained at a pH as low as 5.8 while the extended crystals were obtained at a pH as high as 8. However, this variation in pH should not affect the ionization state of the arginine, aspartic acid, glutamic acid and lysine side chains in the peptide which should maintain an overall net charge of +1 between pH 5.8 and pH 8. In addition, none of the other residues involved in interactions that stabilize the different linker peptide conformations should be affected by the pH range used.

Chameleon sequences are usually recognised as those that adopt different secondary structure when found in different overall protein sequences. In the case of the PmScsC linker, however, the protein sequence is the same and it is the quaternary intermolecular interactions in an assembly of molecules which are promoting the different conformations observed for the linker peptide.

The ability of this shape-shifting motif to adopt α-helix, β-strand and loop conformations is reminiscent of ‘one-to-many’ molecular recognition features (MoRFs) found in intrinsically disordered proteins. These are segments of a disordered protein that are able to adopt different conformations when they undergo ‘disorder-to-order’ transitions upon interaction with multiple binding partners [7,8,9,10]. For example, residues 374–388 in the C-terminus of p53 are disordered in the absence of any binding partners but upon interaction with cyclin A2, sirtuin, CREB binding protein and S100ββ this region adopts a coil, a β-strand, a coil and an α-helix structure, respectively [10].

The β-strand conformation formed in molecule B in the transitional crystal structure (Figure 1c) has relevance for our understanding of protein misfolding and in particular protein aggregation and amyloid formation. This is a clear demonstration of a system where intermolecular interactions, involving both the protein backbone and also charged and polar side chains in hydrogen bond and salt bridge formation, together with van der Waals interactions between more hydrophobic atoms, can stabilise very different conformational properties in a given protein sequence and lead to the formation of short regions of β-sheet. Short sequences, such as that of the linker peptide, in an overall protein sequence could then provide nucleation sites for the overall conversion of a soluble folded protein into amyloid aggregates. Indeed, previous studies have shown that even proteins with a highly helical native structure, such as myoglobin and cytochrome *c*_552_ [31,32], can form amyloid fibrils when the native state is destabilized but conditions favour non-covalent interactions such as hydrogen bonds [33].

In the extended crystal structure, the linker peptide adopts a helical conformation connecting the stalk and catalytic domains (Figure 1a). The observation of a long helix connecting two domains in a crystal structure which converts into a flexible loop region in solution, as reported here, has also been seen in a number of other systems. For example, the crystal structure of calcium-bound calmodulin showed two domains linked by a long, rigid central helix [34,35]. However, NMR studies demonstrated that in solution the central helix is disrupted near its midpoint giving a flexible linker between the two domains [36]. Disruption of a long α-helix connecting two domains in a crystal structure is also seen for the virulence factor Mip protein in solution [37]. Another example is the ribosomal protein L12 where crystallographic studies identified two different dimerization modes [38]. This ribosomal system also has a hinge interdomain linker which is unstructured in solution but helical in the crystal structures [39,40] as seen in PmScsC.

In all of these examples, the flexibility of the region identified as a linker or hinge in solution has been recognised to have functional importance [41]. These regions not only give independent motion to the different domains they connect but also allow for the domains to bind to a variety of different target peptides, proteins or substrates [37,42]. In the case of PmScsC, the flexibility may be important in allowing bound misfolded substrate to access a broad folding landscape so allowing for disulphide bond shuffling [11]. This is in agreement with previously published experimental observations that the deletion or replacement of the linker peptide with a rigid helical peptide abolishes the disulphide isomerase activity of PmScsC [12]. Interestingly, sequence comparisons have identified two other DsbA-like bacterial disulphide isomerases, *L. pneumophilia* DsbA2 and *C. crescentus* ScsC, that contain a peptide sequence that may be indicative of intrinsic disorder in between their N-terminal oligomerization and C-terminal catalytic domains [43]. Thus, it appears that the intrinsic disorder propensity of the PmScsC linker peptide, rather than its ability to adopt different rigid structures, is of key functional importance in PmScsC and potentially in other related bacterial disulphide isomerases.

## Figures and Tables

**Figure 1 biomolecules-11-00642-f001:**
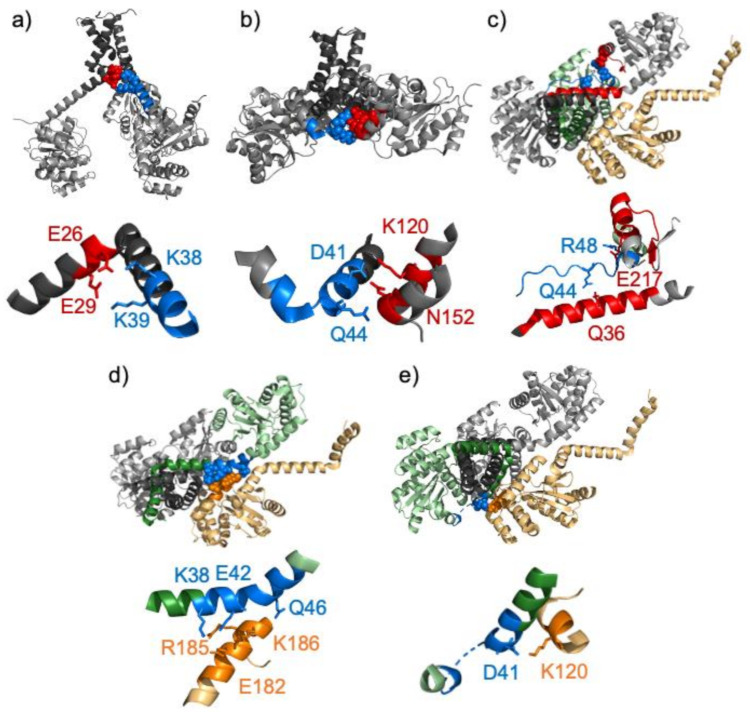
Crystal structures of the PmScsC trimer show several different relative orientations of the trimerization stalk and the catalytic domains and different conformations for the eleven-residue linker peptide. The three molecules which form the trimer are shown in grey with the trimerization stalk shown in a darker grey. Contacts between molecules involving residues within the linker peptide region are shown in the full protein structures and in zoomed-in expansions with important residues shown as spheres and sticks, respectively. (**a**) The extended crystal structure (5id4). The linker peptide sequence is shown in blue in molecule A. Lys 38 and Lys39 make contacts to Glu 26 and Glu 29 in molecule B which are shown in red. (**b**) The compact crystal structure (4xvw). Molecules A, B and F are shown. The linker peptide sequence is shown in blue in molecule A. Asp 41 and Gln 44, shown in blue, make contacts to Lys 120 and Asn 152 in molecule F, shown in red. (**c**–**e**) The transitional crystal structure (5idr). Molecules A, B and C which form a trimer are shown; for clarity, molecules A, B or C are shown in pale green (with the trimerization stalk shown in dark green) in panels d, c or e, respectively. Molecule D from a neighbouring trimer is shown in yellow. In panel c the linker peptide sequence in molecule B is shown in blue and residues 25–44 and 200–217 in molecule A, with which it makes intermolecular contacts, are shown in red. Hydrogen bonds are formed between Gln 44 and Arg 48 on molecule B and Gln 36 and Glu 217 in molecule A. In panel d the linker peptide sequence in molecule A is shown in blue with Lys 38, Glu 42 and Gln 46 highlighted. These three residues make hydrogen bonds to Glu 182, Arg 185 and Lys 186 in molecule D which are shown in orange. In panel e the linker peptide sequence in molecule C is shown in blue and residues 117–120 in molecule D with which it makes intermolecular contacts are shown in orange. There is a salt bridge between Asp 41 in chain C and Lys 120 in molecule D. Note that the coordinates of residues 46–47 are missing from chain C.

**Figure 2 biomolecules-11-00642-f002:**
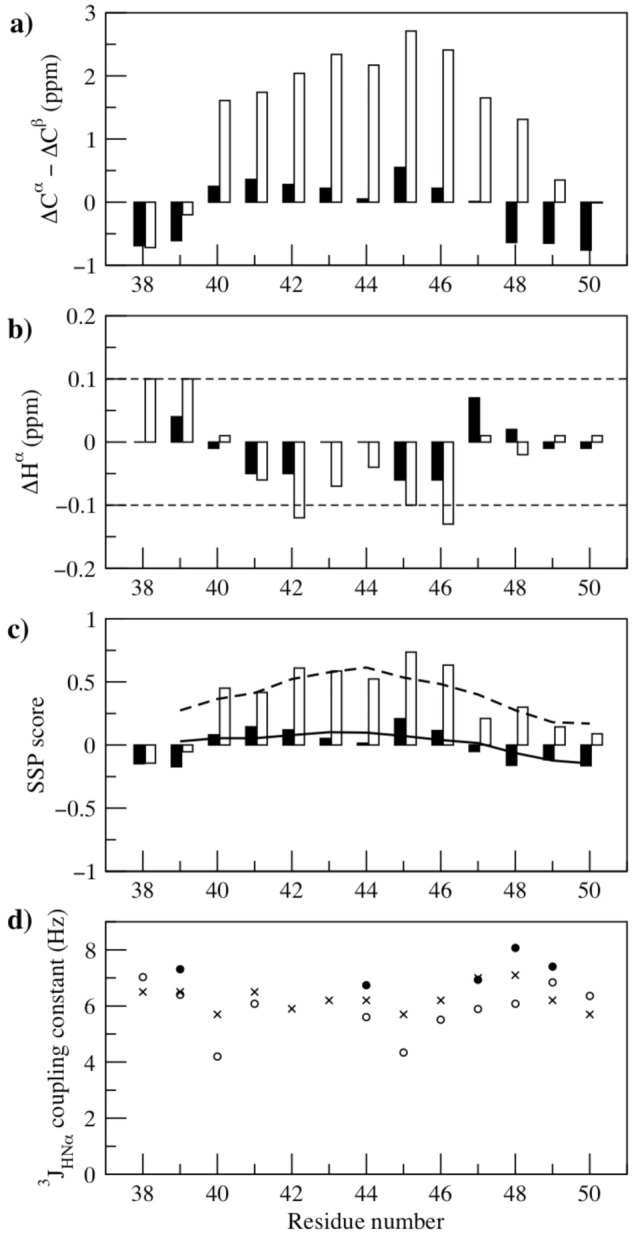
Summary of the NMR data for the linker peptide. (**a**) Secondary shifts ΔC^α^-ΔC^β^. (**b**) Secondary shifts ΔH^α^. (**c**) The Secondary Structure Propensities (SSP) scores. (**d**) The ^3^J_HNα_ coupling constant values. In each panel the data for the linker peptide in water and in 50% TFE solution are shown with a filled and open bar or symbol, respectively. In panel b the dashed lines indicate the 0.1 ppm cut off usually used for secondary structure identification. In panel c the bars show the SSP scores without averaging and the solid and dashed lines show the SSP scores averaged over a 5-residue window for the peptide in water and 50% TFE solution, respectively. The predicted coupling constant values for a random coil, taking into account the character of the preceding residue, are indicated with an x symbol in panel d [26].

**Figure 3 biomolecules-11-00642-f003:**
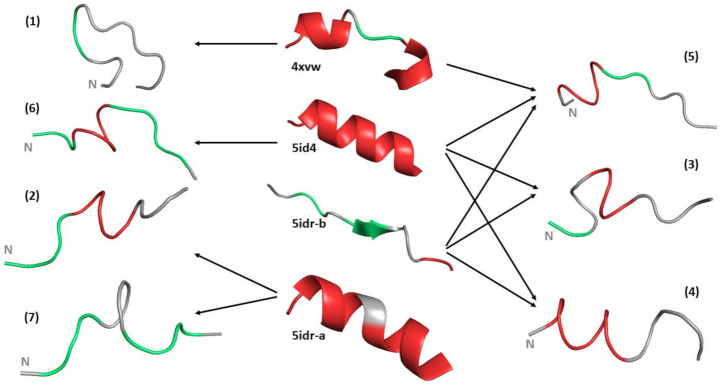
The four central structures show the loop (4xvw, molecule A), helix (5id4), β-strand (5idr molecule B) and irregular helix (5idr molecule A) conformations of the linker peptide in crystal structures which were used as starting structures for the MD simulations. The secondary structure indicated is that present in the intact protein trimer. Structures 1–7 show the conformations at the centres of the seven main clusters identified in the combined four 100 ns MD simulations of the linker peptides. The arrows indicate the main trajectories whose conformers contribute to the cluster. Two or more adjacent residues with Φ, Ψ torsion angles in the α or β region of the Ramachandran plot are shown in red and green, respectively. In each case the peptide is orientated with the N-terminus to the left.

**Figure 4 biomolecules-11-00642-f004:**
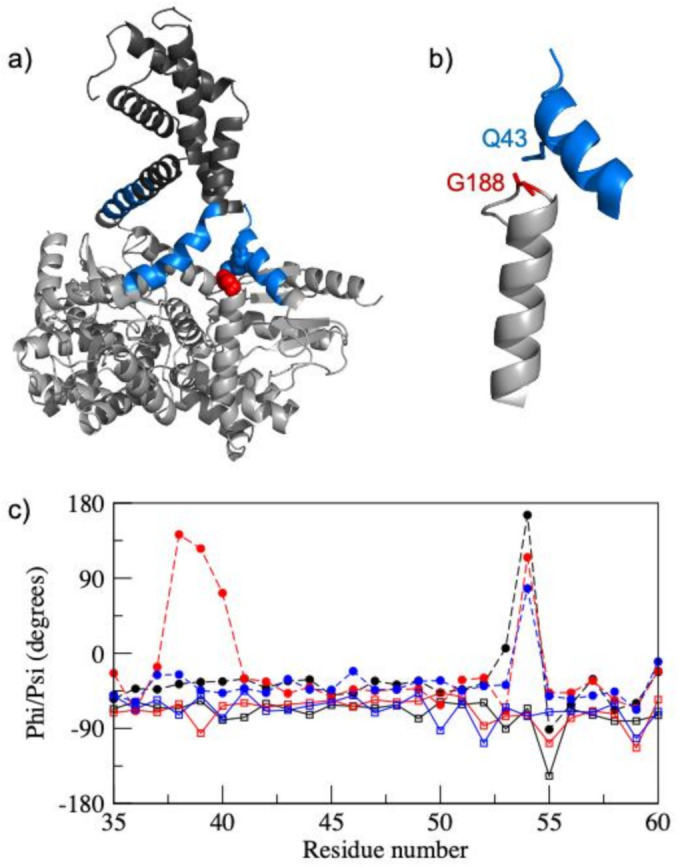
Structural changes observed in a 20 ns molecular dynamics simulation of the extended structure trimer of PmScsC. (**a**) Cartoon representation of the trimer after 20 ns MD simulation. The linker peptide is shown in blue in all three molecules. Gln 43 and Gly 188 from molecule B are shown in blue and red, respectively. (**b**) An expansion showing residues 38–50 and 174–193 from molecule B. Following partial unfolding of the helical linker peptide, the side chain Nε2 of Gln 43 and the backbone CO of Gly 188 form a hydrogen bond which is not found in the starting structure. (**c**) Phi (open square and solid line) and psi (filled circle and dotted line) torsion angles for residues 35–60 in chains A (black), B (red) and C (blue) in the trimer at the end of the 20 ns MD simulation. Residues 38 and 39 adopt extended β-strand phi/psi angles in chain B after the 20 ns simulation.

**Table 1 biomolecules-11-00642-t001:** Hydrogen bonds in the linker peptide in the X-ray structures used as starting coordinate sets for the MD simulations and their percentage populations in the combined trajectory of the four 100 ns peptide MD simulations. Only hydrogen bonds in the X-ray structures, or with a population of at least 20% in the simulations, are included. Note that residue 201 is in the catalytic domain, in the region that forms the β-sheet with the linker peptide in the transitional X-ray structure.

Hydrogen Bond	5idr_A	5idr_B	5id4	4vxw	Trajectory (%)
40 NH-38 O		x			1.5
40 NH-49 O					20.0
42 NH-38 O	x		x	x	5.5
42 NH-39 O					20.0
43 NH-39 O	x		x	x	16.4
44 NH-40 O			x	x	19.2
44 NH-201 O		x			-
45 NH-41 O			x		33.8
45 NH-42 O	x				6.6
46 NHE22-40 O	x				8.1
46 NH-42 O			x		36.1
46 NH-46 OE1		x			4.9
47 NH-43 O			x		29.3
48 NH-44 O	x		x		20.4
48 NH11-41 OD1	x				0.1
49 NH-45 O	x				17.0
50 NH-46 O	x		x		8.1
50 NH-47 O	x		x		2.1
201 NH-44 O		x			-

## Data Availability

The NMR resonance assignments are openly available in the BioMagRes Bank with deposition codes 50399 and 50400.

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
