# Peer review of "The ‘Shape-Shifter’ Peptide from the Disulphide Isomerase PmScsC Shows Context-Dependent Conformational Preferences"

_biomolecules, 2021, doi:10.3390/biom11050642_

Round 1

Reviewer 1 Report

In this manuscript, Smith and co-workers report the structural characterization of a « shape shifter » peptide. It consists of a protein region which can adopt helical, beta-strand and loop conformations. It has been identified in the linker between the trimerization stalk and catalytic domain of the disulphide isomerase PmScsC. The quaternary structure properties adopted by this linker mediate its activity and are context-dependent. In this study, the authors use NMR spectroscopy and molecular dynamics simulations to characterize this linker peptide in solution. This folding study highlights the important role that quaternary interactions can play in determining conformational properties and the impact on protein function but also on misfolding processes such amyloid formation.

In my opinion, the manuscript is clear concise and well written. Figures are coherent with the results they illustrate and importantly the experiments are nicely undertaken. For these reasons, I would recommend publication in this special issue of Biomolecules. I just have minor points.

1) The authors mention some CD experiments in presence of TFE. It would be nice to add a figure as supplementary materials.

2) This is just a question for my own interest: 

Is this peptide sequence common in many proteins? 

For example, what is its conservation degree in the thioredoxin-fold family?

Author Response

  1. As requested, we have added a Supplementary Figure (S3) showing the CD spectra and have added a reference to this on line 206.
  2. Some DsbA-like proteins with thioredoxin-fold domains have N-terminal extensions to the sequence which are responsible for oligomerisation. Some of these contain what have been suggested to be ‘shape-shifter’ peptides between the N-terminal oligomerization and the thioredoxin-fold domains. These peptides do not show significant sequence homology but contain residues such at Gln which are associated with disordered structure; these sequences may provide the flexibility required for the isomerase activity. We have added a sentence about this to our Discussion (see lines 382-388 and new reference 43)

Reviewer 2 Report

In this study, the authors done the conformational characterization of the linker peptide of Proteus mirabilis suppressor of copper sensitivity protein C (PmScsC). The solution structure of the peptide was investigated by NMR and molecular dynamics simulations. Experiments including the simulation were carefully done, and the paper was well written. The reviewer has only minor comment. The representative structures of seven main clusters observed in MD simulation are shown in Figure 3. The structure 2 and 3 look the same. Are they correct?

Author Response

  1. We thank the reviewer for pointing out the error in Figure 3. We have replaced structure 2 with the correct structure in the revised Figure 3.

Reviewer 3 Report

The authors analyzed the chemical shifts and secondary structures of the PmScsC linker region, which undergoes conformation changes between the disordered and helical structures. This study may further the understanding of one-to-many molecular recognition features of IDPs. The following concerns should be addressed.

  1. Mutagenesis studies or others may be included to support the MD simulation results.
  2. L49-54 & Figure 1, the orders of the figures should be consistent with the main text.
  3. L176, the 1H-15N HSQC spectrum should also be included in Figure S1.

Author Response

  1. Furlong et al 2017, 2019 report studies of variants of PmScsC in which the linker peptide sequence is deleted and in which the linker peptide is replaced by a rigid helical peptide. In both these mutational studies the disulphide isomerase activity of the protein is lost. In addition, small angle scattering data show that the variant of PmScsC with the rigid helical peptide has reduced flexibility. These results are consistent with the data reported in our manuscript. Moreover they show that the high flexibility of the linker peptide that we report is functionally important. This is discussed on lines 60-68 and 380-382 in our manuscript.  
  2. We have edited the text on lines 50 to 53 so that Figure 1a is mentioned before Figure 1b. In Figure 1 we have swapped panels c and d so that on line 57 we mention Figure 1c rather than Figure 1d.
  3. As requested, we have added a Supplementary Figure (S2) showing the 1H-15N HSQC spectra in H2O and in 50% TFE and have added a reference to this on line 179.

Reviewer 4 Report

The authors characterized a “shape-shifter” peptide derived from disulphide isomerase PmScsC using NMR experiments and MD simulations. This peptide is known to be essential for the enzyme activity from previous studies. The main findings from the current study are that the peptide is disordered in solution and the various secondary structures (a-helix, b-strand and random coil) of this peptide in the context of the entire trimeric protein result from intersubunit contacts. The NMR data confirmed the random coil structure of this peptide in solution. The MD simulations and analysis of crystal structure revealed H-bonds that are relatively short-lived, highlighting the conformational flexibility of the peptide even in the context of the full-length protein. The results were clearly described and the manuscript was very well written. I suggest publication of this manuscript after minor revision. Below are my specific suggestions for minor changes.

  1. In the introduction, the authors mentioned three different crystal structures of PmScsC trimer. One of the three structures was obtained under a different crystallization condition. This fact should be mentioned to emphasize that the observed conformations may depend on conditions (especially pH). In the discussion section, I suggest that the effects of pH and other conditions on peptide conformations should be discussed in general and in the context of this particular enzyme.
  2. For figure 1 panels c)-e), I suggest that molecules A, B, C be highlighted in a different color (other than grey) in the corresponding panels (i.e. molecule A for panel c), B for panel d) and C for panel e).
  3. In line 220, there should be a space between the number and the unit in “100ns”. The same change should be made throughout the entire text whenever the space is missing.
  4. Line 222, “figure 3 centre” should be changed to “figure 3 the column in the center” for clarity
  5. Line 339, figure 1 should be cited since molecule B in one of crystal structures is mentioned in this sentence.

Author Response

  1. We have added a sentence to the Introduction that explains that the crystals were obtained under different conditions (particularly pH) (lines 44-47). In the Discussion section we also now discuss the different pH values in the context of the interactions that the peptide makes within the different structures (lines 328-334). We expect that in the pH range of 5.8 to 8, the ionization state of the Arg/Asp/Glu/Lys sides chains in the linker peptide will not change and that the overall charge of the peptide will be +1. Since none of the stabilizing interactions involve His, we believe that these will also not be sensitive to this pH range.
  2. In order to distinguish the three molecules making up the trimers shown in Figures 1 c-e, we have highlighted the particular monomer of interest in each panel using a light green (with darker green for the trimerization stalk). We hope this makes Figure 1 easier to interpret.
  3. We have added a space between the number and the unit in ‘100 ns’ and at other relevant places in the manuscript (including units of ns, ps, nm, ppm etc).
  4. We have changed the reference to ‘Figure 3 centre’ to ‘Figure 3 column in the centre’ as suggested (line 224-225).
  5. We have added a reference to Figure 1 at line 349 where we mention the transitional structure. In addition, a reference to Figure 1 has also been added at line 362 where we mention the extended structure.

Round 2

Reviewer 3 Report

The authors have addressed my concerns.